# Natural variation in temperature-modulated immunity uncovers transcription factor bHLH059 as a thermoresponsive regulator in *Arabidopsis thaliana*

**Friederike Bruessow**[1,2], **Jaqueline Bautor**[1], **Gesa Hoffmann**[1¤], **Ipek Yildiz**[3], **Jürgen Zeier**[2,3], **Jane E. Parker**[1,2]*

1 Department of Plant-Microbe Interactions, Max-Planck Institute for Plant Breeding Research, Cologne, Germany, 2 Cologne-Düsseldorf Cluster of Excellence on Plant Sciences (CEPLAS), Düsseldorf, Germany, 3 Institute of Plant Molecular Ecophysiology, Heinrich Heine University, Düsseldorf, Germany

¤ Current address: Department of Plant Biology, Uppsala BioCenter, Swedish University of Agricultural Sciences (SLU), Uppsala, Sweden
* parker@mpipz.mpg.de

**Data Availability Statement:** All data are in the manuscript and its supporting information files.

## Abstract

Temperature impacts plant immunity and growth but how temperature intersects with endogenous pathways to shape natural variation remains unclear. Here we uncover variation between *Arabidopsis thaliana* natural accessions in response to two non-stress temperatures (22˚C and 16˚C) affecting accumulation of the thermoresponsive stress hormone salicylic acid (SA) and plant growth. Analysis of differentially responding *A. thaliana* accessions shows that pre-existing SA provides a benefit in limiting infection by *Pseudomonas syringae* pathovar *tomato* DC3000 bacteria at both temperatures. Several *A. thaliana* genotypes display a capacity to mitigate negative effects of high SA on growth, indicating within-species plasticity in SA—growth tradeoffs. An association study of temperature x SA variation, followed by physiological and immunity phenotyping of mutant and over-expression lines, identifies the transcription factor *bHLH059* as a temperature-responsive SA immunity regulator. Here we reveal previously untapped diversity in plant responses to temperature and a way forward in understanding the genetic architecture of plant adaptation to changing environments.

## Author summary

Temperature has a profound effect on plant innate immune responses but little is known about the mechanisms underlying natural variation in transmission of temperature signals to defence pathways. Much of our understanding of temperature effects on plant immunity and tradeoffs between activated defences and growth has come from analysis of the common *Arabidopsis thaliana* genetic accession, Col-0. Here we examine *A. thaliana* genetic variation in response to temperature (within the non-stress range—22 ˚C and 16 ˚C) at the level of accumulation of the thermoresponsive biotic stress hormone salicylic

**Funding:** The study was financed by the Cluster of Excellence on Plant Science grant EXC-2048/1, Project 390686111 (www.ceplas.eu) and F.B. was supported by the Swiss National Science Foundation grant PBLAP3-142776 (www.snf.ch). The funders had no role in study design, data collection and analysis, decision to publish, or preparation of the manuscript.

**Competing interests:** The authors have declared that no competing interests exist

acid (SA), bacterial pathogen resistance, and plant biomass. From analysis of 105 genetically diverse *A. thaliana* accessions we uncover plasticity in temperature-modulated SA homeostasis and in the relationship between SA levels and plant growth. We find that high SA amounts prior to infection provide a robust benefit of enhancing bacterial resistance. In some accessions this benefit comes without compromised plant growth, suggestive of altered defence–growth tradeoffs. Based on a temperature x SA association study we identify the transcription factor gene, *bHLH059*, and show that it has features of a temperature-sensitive immunity regulator that are unrelated to *PIF4*, a known thermosensitive coordinator of immunity and growth.

## Introduction

Analysis of phenotypic variation is a means to identify genes and networks underlying complex traits[1]. Environment shapes plant phenotypes and is a driver of adaptation to new habitats [2–4]. Temperature, as one key environmental variable, impacts plant physiology, growth and responses to abiotic and biotic stresses[5,6]. As temperature fluctuations across the globe increase, it is important to determine how plants integrate temperature signals with plant developmental and stress programs, and the genetic networks enabling resilience to climate change.

There has been recent progress in elucidating processes that coordinate temperature with plant endogenous pathways. Phytochromes act as thermosensors, coupled with their central integrative role in light quality perception and signalling[7–9]. In *Arabidopsis thaliana*, phytochrome B (phyB) regulates the bHLH transcription factor phytochrome interaction factor 4 (PIF4) to prioritize growth over immune responses at elevated temperatures via the De-Etiolated 1 (DET1) and Constitutive Photomorphogenic 1 (COP1) photomorphogenic regulatory module [10,11]. Also, sumoylation events mediatated by SUMO E3 ligase SIZ1 affect COP1/ PIF4-dependent growth-defense tradoffs at high temperatures [12]. In the cold, membrane-bound NAC transcription factor NTL6 is released to induce disease resistance[13]. The Early Flowering 3 (ELF3) scaffold protein for a temperature-responsive transcription repressor evening complex has a directly thermosensitive prion-like domain [14]. Therefore, temperature signals influence transcriptional regulation of immunity and growth.

Coordination between temperature and plant resistance to pathogen infection is determined by phytohormone pathways with contrasting roles in growth and defence, and by temperature effects on *in planta* microbial metabolism and infectivity[6,15–18]. In *A. thaliana*, the two major protective layers against microbial pathogens: cell surface-based pattern-triggered immunity (PTI) and intracellular effector-triggered immunity (ETI) respond differently to ambient temperature, with PTI being preferentially activated at elevated and ETI at lower temperatures[17,19,20]. Heat stress events can, however, drastically reduce PTI responsiveness [21]. Gradual depletion of the alternative histone H2A.Z in nucleosomes with increasing temperature[22] is associated with increased PTI-dependent gene expression at the expense of ETI [19]. Hence, temperature effects registered at the chromatin are also important for plant immunity outputs.

The plant stress hormone, salicylic acid (SA), mediates basal and systemic immunity to biotrophic and hemi-biotrophic pathogens by reprogramming cells for defence via the transcriptional co-regulator, nonexpressor of *PR1* (NPR1) [23]. In *A. thaliana*, pathogen-induced SA is generated mainly by the isochorismate synthase1 (ICS1) pathway[24,25]. Induction of *ICS1* expression and pathogen resistance in *A. thaliana* basal and ETI responses are compromised

at temperatures above 23–24 °C [5,16,26]. In *A. thaliana* accession Col-0, increased SA was also responsible for plant stunting after a shift from 23˚C to near chilling conditions (5˚C) [27]. Exposure of other *A. thaliana* accessions to 10˚C revealed genotype-specific expression patterns for ~75% cold-regulated transcripts[28], highlighting the extent of natural variation in temperature-modulated gene expression. Additionally, SA is an inducer of thermogenesis in certain plant species[29], broadening its role in temperature responses.

Lower temperatures (<16˚C) amplify pathogen-activated ETI and autoimmune responses (- the latter often due to mis-activated ETI receptors) accompanied by increased SA production and pathogen resistance[20,26,30–34]. Plant autoimmune backgrounds exhibit stunting and leaf necrosis as negative consequences of activated defences on plant fitness[5]. Defence—growth tradeoffs appear to be hard-wired through phytohormone and transcriptional networks[35–38], probably to steer the plant through stressful periods[39,40]. However, there are instances in which antagonistic interactions between stress and growth pathways are uncoupled [38,40–44], indicating genotypic and phenotypic plasticity in defence—growth coordination.

Here we investigate *A. thaliana* natural genetic variation in immunity and growth responses to two temperature regimes (22/20˚C and 16/14˚C). Our aim was to assess the phenotypic space in immunity x growth interactions over a non-stress temperature range for this species. Using SA accumulation in leaves of 105 genetically diverse accessions as a first proxy for defence homeostasis, we uncover variation in temperature modulation of SA and in the relationship between leaf SA and biomass. At both temperatures, there is a measurable benefit of high initial SA levels on plant resistance to a leaf-infecting bacterial pathogen, *Pseudomonas syringae* pathovar *tomato* DC3000 (*Pst* DC3000), after it has passed a stomatal barrier (post-stomatal resistance). A genome-wide association study of temperature x SA variation identifies the bHLH059 transcription factor as a new thermoresponsive immunity component.

## Results

### SA chemotyping of *A. thaliana* plants at two temperature regimes

To measure temperature-modulated SA accumulation we selected 105 *A. thaliana* accessions from the HapMap population based on genetic diversity and geographical distance[45]. Most accessions (80%) originate from Eurasia populations and we included naturalized lines from America, Africa, New Zealand and Japan (S1 Table). Individual plants were grown in separate pots to avoid competition/shading and, as a randomized design in controlled cabinets, kept at 16˚C/14˚C or 22˚C/20˚C and 12 h light/dark cycle within the non-stress range for *A. thaliana* [46]. We then determined biomass and SA contents of 5-week-old plants under each temperature regime. Because there was a strong correlation between fresh and dry plant weights (S2 Table), we used above-ground fresh weight (FW) as a measure of biomass.

To quantify SA in a large number of samples, we used a high-throughput SA biosensor-based luminescence method[47,48] (Methods). This provided total SA measurements in medium (Ws-0 and Col-0) and high SA (C24 and Est-1) accumulating accessions[41,49] with an accuracy comparable to GC-MS (S1A Fig). The biosensor method was less reliable for quantifying low levels of free SA, the biologically active form (S1B Fig) [50]. There was a high correlation between free and total SA amounts in GC-MS assays of 15 tested 5-week-old accessions with contrasting SA contents at 22˚C (S2 Fig). We therefore used biosensor-based total SA as a measure of SA accumulation at the two temperatures. As plant age influences SA accumulation and outputs[40], we assessed whether differential SA accumulation between accessions is captured reliably at 5 weeks. For this, total SA was quantified in five accessions which in pilot studies had shown low (Sha, Col-0), intermediate (Est-1) or high (An-1) total SA

contents, together with a Col-0 *isochorismate synthase* SA biosynthesis mutant *sid2-1*[24], over a 7-week time course. Total SA accumulation trends seen in 5-week-old plants persisted over the course of development from 4–7 weeks regardless of flowering time (S3 Fig).

### Genetic variation in *A. thaliana* SA—growth tradeoffs

At each temperature there was considerable genetic variation in plant biomass and total SA levels between accessions (Fig 1A and 1B and S1 Table). Surprisingly, total SA did not show a general tendency to increase in plants grown at 16˚C compared to 22˚C, although biomass at 16˚C was lower (Fig 1A–1C). Therefore, increased SA at cooler temperatures reported previously for accession Col-0[13,27], and also found here for Col-0 (S1 Table), appears not to be generalizable for *A. thaliana*. Moreover, comparing total SA contents with biomass in each accession revealed that at 16˚C and 22˚C there was an extremely weak negative correlation between total SA levels and above ground FWs (Fig 2A and 2B). One third of accessions with total SA contents >1 µg/g FW had a biomass above the median at each temperature, suggesting that there is within-species genetic and/or phenotypic plasticity in SA—growth tradeoffs.

Negative effects of defence on growth in high SA backgrounds might be mitigated by imposing a higher induction threshold for SA immunity. We therefore measured expression of the SA-responsive *pathogenesis-related 1* gene (*PR1*) at 22˚C in selected 5-week-old accessions with high total SA amounts (>1 µg/g FW) and high biomass (>1.5 g) (Ven-1, PHW-13, Kas-2), accessions with high total SA and low biomass (>0.5 g) and some leaf necrosis (Gy-0, Spr1-2, Mz-0), or with low total SA (<0.3 µg/g FW) and varied biomass (Mrk-0, Col-0, Oy-0) (Fig 2C–2F). For these nine tested accessions, high total SA was accompanied by elevated *PR1* expression but not always stunting (Fig 2C–2F). The data suggest that mechanisms other than responsiveness to SA reduce antagonism of growth in some *A. thaliana* genetic backgrounds. Whether natural variation in SA x growth tradeoff at 22˚C also involves cold-regulated mechanisms of growth reduction such as phytochrome signalling [7], auxin transport [51] and/or gibberellic acid and inhibitory DELLA interplay [52,53], remains unclear.

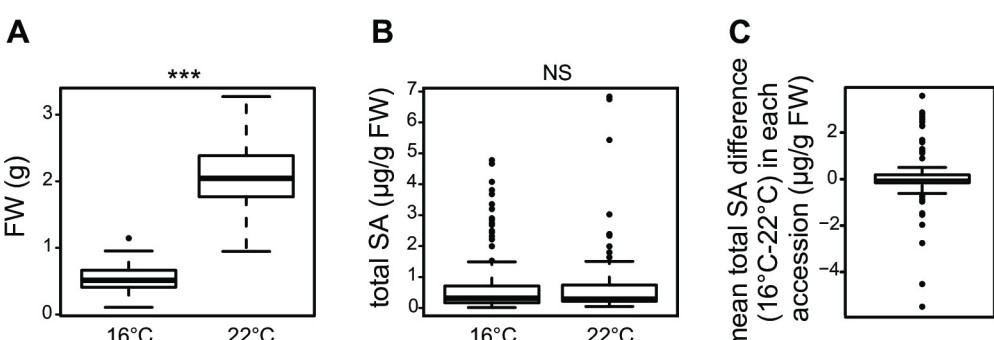

**Fig 1. Analysis of biomass and total SA levels in 105 *A. thaliana* accessions grown for 5 weeks at two temperature regimes (22˚C and 16˚C) reveals natural variation in SA homeostasis in response to temperature.** Data are represented as boxplots. **A)** Aboveground fresh weight (FW) of plants grown at 16˚C (n = 420, four biological replicates) and 22˚C (n = 315, three biological replicates). Statistical difference according to kruskal-wallis non parametric test with p-value<0.05 is indicated above the plot with a star **B)** Leaf total SA contents of plants grown at 16˚C (n = 420) and 22˚C (n = 315). Statistical difference according to kruskal-wallis non parametric test is indicated above the plot. NS = not significant. **C)** Distribution of mean leaf total SA differences in accessions between temperature regimes (n = 105). Accessions with higher SA contents at 16˚C than 22˚ are at the positive side and accessions with higher SA contents at 22˚C than at 16˚C on the negative side of the plot. Accessions with little or no change in SA contents in response to temperature score around 0.

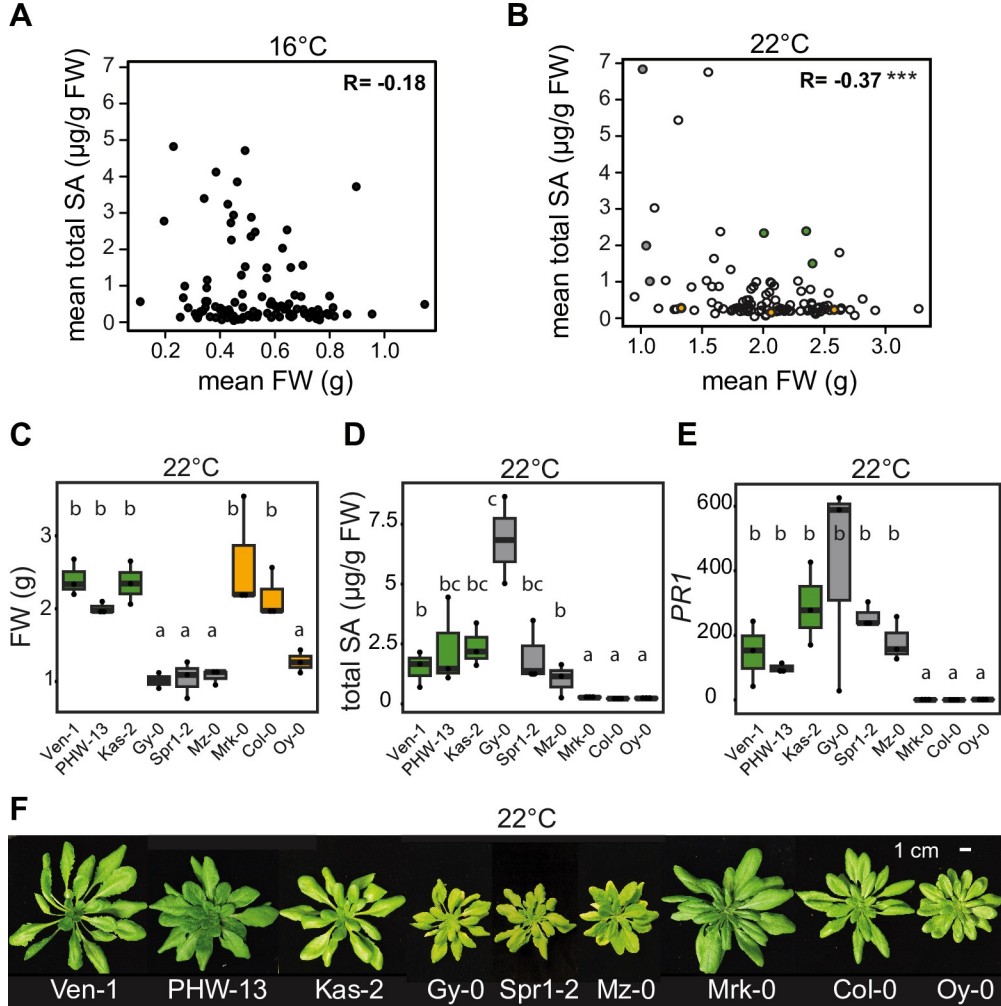

**Fig 2. Natural variation in growth—defense tradeoffs in *A. thaliana*. A)** Scatterplot of mean above-ground fresh weight (FW) according to mean leaf total SA contents of 105 *A. thaliana* accessions grown for 5 weeks at 16˚C. R = Pearson's correlation index. Data are from four biological replicates (t = -1.9184, df = 103, p-value = 0.05783). **B)** Scatterplot of mean above-ground FW according to mean leaf total SA contents of 105 *A. thaliana* accessions grown for 5 weeks at 22˚C. R = Pearson's correlation index. Data are from three biological replicates (t = -4.0581, df = 103, p-value = 9.651e-05). Green dots represent accessions with high biomass and high SA accumulation, grey dots represent accessions with low biomass and high SA accumulation and orange dots represent accessions with varied biomass and low SA accumulation phenotyped in Fig 2C–2F. **C)** Above-ground FW in 5-week-old plants of nine *A. thaliana* accessions grown at 22˚C (n = 3 biological replicates). Letters indicate significant differences after FDR multiple testing correction in one-way ANOVA. **D)** Leaf total SA contents in 5-week-old plants of nine *A. thaliana* accessions grown at 22˚C (n = 3 biological replicates). Letters indicate significant differences after FDR multiple testing correction in one-way ANOVA. Data was log(10) transformed for statistical analysis. **E)** Expression of *PR1* relative to *SAND* reference gene in 5-week-old plants of 9 *A. thaliana* accessions grown at 22˚C (n = 3 biological replicates). Letters indicate significant differences after FDR multiple testing correction in one-way ANOVA. Data was log(10) transformed for statistical analysis. **F)** Visual growth phenotypes of 5-week old *A. thaliana* accessions examined in c) to e).

## SA accumulation in response to temperature is genotype-specific

Because SA amounts in leaves of different accessions did not relate strongly with reduced growth, we examined the effect of temperature on SA homeostasis regardless of biomass. From the initial 105 *A. thaliana* accessions, we selected lines that accumulated higher total SA at 22˚C than 16˚C (Ven-1, Mz-0, Nok-3), lines that showed no variation in total SA levels between the two temperatures (Se-0, NFA-8), and lines displaying higher total SA at 16˚C than

at 22˚C (Fei-0, Ei-0, Bay-0, Est-1) (S1 Table and Fig 3A). Reference accessions Col-0 and Ler-0 had relatively small but opposing total SA responses to temperature (S1 Table and Fig 3A). We concluded that even a moderate change in temperature within the normal range of *A. thaliana* (here 6˚C) exposes variation in SA pathway homeostasis. To further validate our findings, we repeated total SA quantitation by GC-MS analysis in Ven-1, Mz-0, Fei-0, Ei-2 and Col-0 grown at 16˚C and 22˚C and observed similar temperature-dependent patterns as with the biosensor-based method (Figs 3A and S4A). Temperature-dependent expression of the SA-responsive gene *PR1* in Ven-1, Mz-0, Fei-0, Ei-2 and Col-0 correlated with total SA contents (S4B Fig), suggesting that different SA contents impact SA-dependent defense gene expression. Because differential SA accumulation in response to temperature might be driven by changes in SA biosynthesis, we measured expression of three SA biosynthesis genes in the same leaf tissues of these accessions. *Isochorismate synthase 1* (*ICS1*) and *avrPphB SUSCEPTIBLE 3* (*PBS3*) are key enzymes for pathogen-induced SA synthesis[25] and *phenylalanine ammonia lyase 4* (*PAL4*) regulates SA basal accumulation [54]. None of these genes showed differential regulation in response to temperature in Ven-1, Mz-0, Fei-0, Ei-2 and Col-0 (S10C–S10E Fig). These data suggest that temperature-dependent SA accumulation within this 16˚C– 22˚C range is not determined by strongly altered expression of SA biosynthesis genes.

## Temperature-modulated SA impacts bacterial pathogen growth in leaves

With differences of up to 5 μg total SA/g FW in accessions grown under the two temperature regimes (Fig 1C), we anticipated temperature-dependent variation in immune responses between accessions, as suggested by the *PR1* expression profiles of selected genotypes at 22˚C (Figs 2E and S4B). We spray-inoculated leaves of 11 5-week-old accessions showing diverse total SA levels at 16˚C and 22˚C (Fig 3A) with virulent *Pst* DC3000 at these two temperatures. *Pst* DC3000 produces the JA-Ile mimic coronatine (COR) which promotes reopening of leaf stomata to counter bacterial PAMP-induced stomatal closure and increase bacterial entry to the leaf apoplast[18]. Host-produced SA promotes stomatal closure and post-stomatal resistance to *Pst* DC3000[18,55,56]. At 3 h post inoculation (hpi), *Pst* DC3000 levels inside leaves were unchanged between temperatures in each accession but showed up to 10-fold differences between accessions (S5A Fig), suggesting that differences in early stomatal entry of bacteria into leaves is not a major variable between 16˚C and 22˚C in these accessions. After measuring *Pst* DC3000 growth in leaves of plants at 4 dpi, we found a robust inverse correlation between temperature-modulated total SA accumulation and bacterial growth across accessions (Fig 3B). Thus, in accessions showing a rise in total SA between 16˚C and 22˚C there was increased resistance to *Pst* DC3000 and the opposite trend was observed in plants which had reduced total SA between 16˚C and 22˚C (Fig 3B). Accessions which responded negligibly to temperature at the level of SA accumulation showed no difference in temperature effects on *Pst* DC3000 infection (Fig 3B). These data reveal a positive relationship between temperature-modulated total SA accumulation and limitation of *Pst* DC3000 growth in leaves of the 11 tested *A. thaliana* accessions.

We observed variation between accessions in the degree to which total SA differences impacts bacterial resistance. For example in Ler-0, a rise of only 0.38 μg/g FW total SA between 16˚C and 22˚C resulted in a substantial (1.5 $\log_{10}$) reduction in bacterial numbers (Fig 3B). Est-1, with a much higher total SA differential (1.68 μg/g FW) between temperatures, showed only a small (0.5 $\log_{10}$) difference in bacterial growth, whereas in Fei-0 a 2.56 μg/g FW total SA change translated to a 2.0 $\log_{10}$ bacterial growth difference (Fig 3B). All accessions in Fig 3 had proportional free and total SA levels (S2 Fig). Together, these data suggest there is variation between *A. thaliana* accessions in the extent to which accumulated SA translates to bacterial

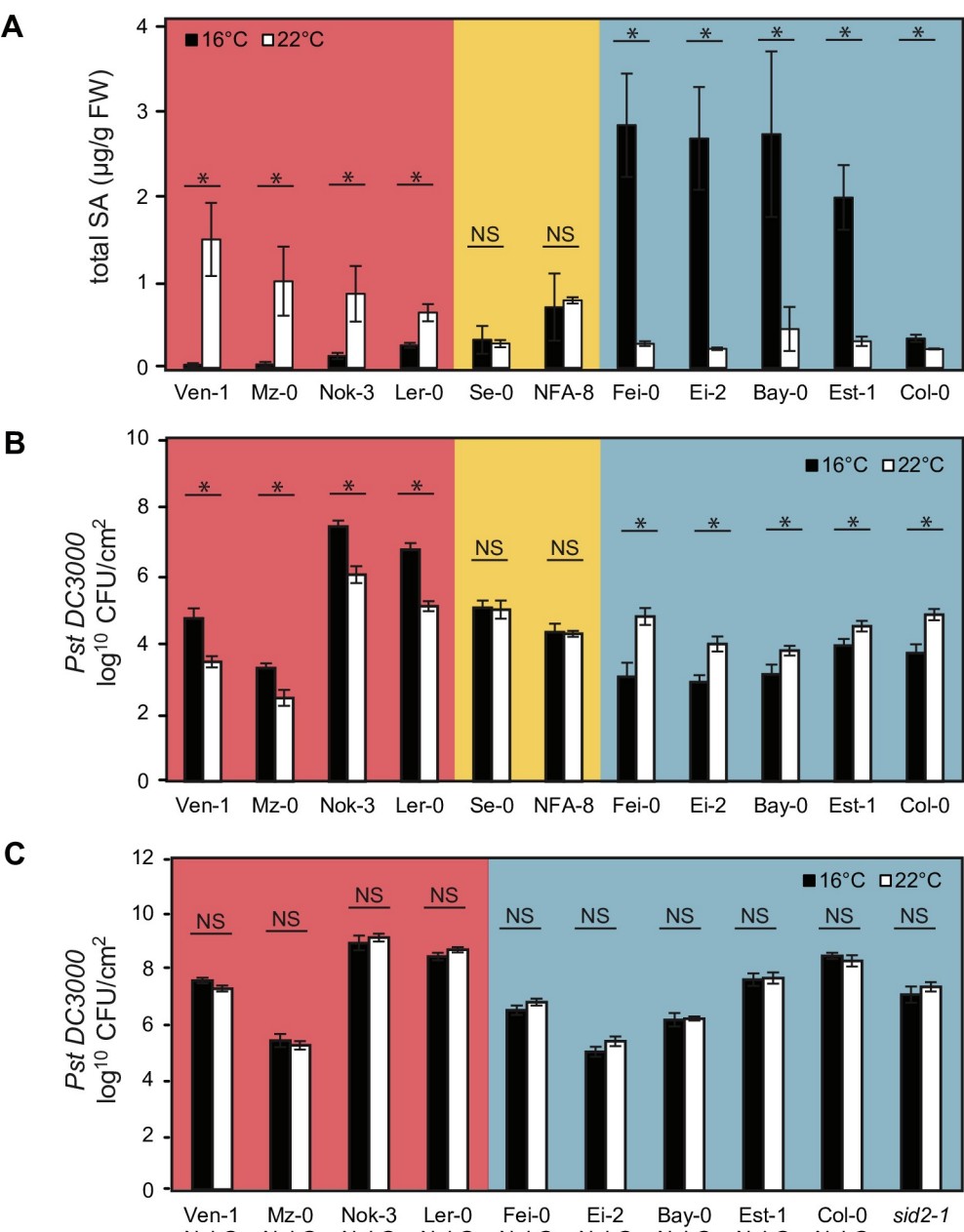

**Fig 3. Temperature-modulated SA accumulation impacts resistance to *Pst* DC3000. A)** Leaf total SA content in 5-week-old plants of 11 *A. thaliana* accessions grown at 16˚C (n = 4 biological replicates) or 22˚C (n = 3 biological replicates). Significant differences between temperature regimes after kruskal-wallis non parametric test with p-value<0.05 are indicated with a star. NS = not significant. Error bars represent standard error. Colour blocks indicate phenogroups **B)** Bacterial titres in leaves of 5-week-old plants of 11 *A. thaliana* accessions grown at 16˚C or 22˚C at 4 dafter spray-inoculation with *Pst* DC3000 (n = 18, three biological replicates). Significant differences between temperatures after Student t-test with p-value<0.05 are indicated on plot with a star. NS = not significant. Error bars represent standard error. Day 0 sample measurements are shown in S5 Fig. **C)** Bacterial counts in leaves of 5-week-old plants of 11 SA-deficient (*NahG* transgenic) *A. thaliana* accessions grown at 16˚C or 22˚C, at 4 d after infection with *Pst* DC3000 (n = 18 except for Ven-1 NahG where n = 12, three biological replicates). Significant differences between temperatures after Student t-test with p-value<0.05 are indicated with a star. NS = not significant. Error bars represent standard error. Day 0 sample measurements are shown in S5 Fig.

immunity. When cultured on liquid M9 minimal salt medium containing sorbitol as carbon source over a 56 h time course, *Pst* DC3000 grew more slowly at 16°C than at 22°C during the exponential phase (S6 Fig). This result emphasizes the influence of *A. thaliana* host genotype in determining temperature effects on bacterial growth in leaves. In order to delineate geno-type-dependent stomatal vs apoplastic defenses at 16°C and 22°C, we syringe-infiltrated *Pst* DC3000 bacteria into leaves of five of the 11 differentially responding genotypes (Ven-1, Mz-0, Se-0, Fei-0 and Col-0) to bypass stomata, and performed a bacterial growth time course over 4 days at the two temperatures (S7A–S7E Fig). We observed similar temperature-dependent bacterial growth outcomes as in the spray inoculated plants (compare S7 Fig with Fig 3B). Therefore, we propose that apoplastic immunity contributes substantially to the observed genetic variation in temperature effects on *Pst* DC3000 infection.

## High SA accumulation prior to infection increases bacterial immunity

We have shown that *A. thaliana* SA amounts before infection correlate positively with resistance to virulent *Pst* DC3000. Next we tested whether temperature effects on *Pst* DC3000-induced SA might also contribute to resistance. For this, leaves of accessions Ven-1, Mz-0, Se-0, Fei-0, Ei-2 and Col-0 grown at 16°C or 22°C were sprayed with *Pst* DC3000 or buffer (mock) and total SA measured at 24 hpi. Similar temperature effects on total SA accumulation were observed in these accessions after mock treatment as in untreated plants (compare Fig 4 (mock) and Fig 3A). After *Pst* DC3000 inoculation, there were no significant temperature differences in total SA accumulation between accessions (Fig 4). Hence, accessions with lower starting (basal) SA at 16°C or 22°C induced SA to comparable levels as high initial SA accumulators at 24 hpi (Fig 4). We concluded that temperature modulated SA accumulation before infection is an important determinant of *A. thaliana* immunity to *Pst* DC3000 bacteria in the 16°C to 22°C range.

## SA underlies differential temperature effects on resistance to bacteria

We tested whether the observed temperature-dependent differences in bacterial resistance between accessions are determined by SA levels. For this, we introduced into different *A. thaliana* accessions a bacterial *NahG* (*salicylate hydroxylase*) gene which breaks down SA to

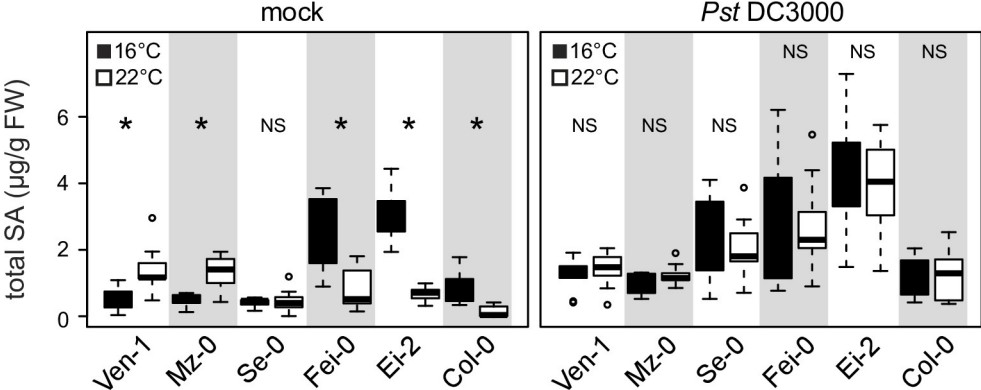

**Fig 4. Inducibility of total SA by *Pst* DC3000 of leaves at 16°C and at 22°C.** Leaf total SA contents in 5-week old *A. thaliana* accessions with different temperature-modulated SA contents were assessed at 24 h after spray-treatment with 10mM MgCl$_2$ (mock) or *Pst* DC3000 (OD$_{600}$ = 0.15). Three individual plants per experiment were assessed per experiment and three independent experiments performed (n = 9). Significant differences between temperatures after Student t-test with p-value<0.05 are marked with a star. NS = not significant.

catechol[57]. A single *NahG* transformant line from each accession was selected after checking total SA depletion at the temperature the parental accession produced highest SA amounts (S8 Fig). The SA-depleted (*NahG*) accessions, together with existing Ler-0 and Col-0 *NahG* lines [58,59] grown at 16˚C or 22˚C, were spray-inoculated with *Pst* DC3000 and bacterial titers measured in leaves at 3h and 4 dpi. There was again no detectable temperature effect on *Pst* DC3000 early stomatal entry to leaves at 3 hpi (S5B Fig). In contrast to the parental responses (Fig 3B), corresponding *NahG* lines had lost temperature-dependent differential resistance to *Pst* DC3000 growth at 4 dpi (Fig 3C). This loss was also observed in Col-0 *sid2-1* (Fig 3C). These data suggest that temperature-regulated SA accumulation directly or indirectly underlies the observed temperature effects on resistance to virulent *Pst* bacteria.

Notably, variation in *Pst* DC3000 growth between wild-type accessions persisted in the corresponding *NahG* transgenic lines that was independent of temperature (Fig 3B and 3C). For example, up to a 1000-fold difference in *Pst* DC3000 titers was observed between the most susceptible (Nok-3 and Ler-0) and resistant (Mz-0 and Ei-2) genotypes (Fig 3C). These data highlight a substantial contribution of SA-independent processes in limiting *Pst* DC3000 growth which, unlike the SA-dependent resistance, are unaffected by changes in temperature within the 16˚C to 22˚C range. We concluded that there is within-species genetic variation in both temperature-dependent SA and temperature-independent (non-SA) defences shaping *A. thaliana* post-stomatal immune responses to bacteria.

## Genetic architecture of SA regulation by temperature in *A. thaliana*

After assessing SA homeostasis in response to temperature in 105 *A. thaliana* accessions, we examined whether specific phenotypes fit a global distribution pattern, using the coefficients of a GLM (Generalized Linear Model; Materials and Methods) to colour-coded phenotypes at occurrence sites. No obvious geographic or climatic distribution patterns were found for temperature-dependent total SA regulation (Methods) (Fig 5A; representing only extended European accessions for clarity).

Broad sense heritability of SA accumulation was calculated to be 0.79 at 16˚C and 0.76 at 22˚C, indicating a sizable genetic underpinning to this trait. To explore the trait genetic architecture we performed temperature x total SA association mapping on 99 accessions using the GWAPP tool[60] and coefficients of the GLM as a phenotype (Methods). One major peak on the upper arm of chromosome 4 contained six significant single nucleotide polymorphisms (SNPs) after Bonferroni multiple testing correction (Fig 5B and S3 Table). Two additional peaks were found on chromosomes 1 and 4, each with one significantly associated SNP after Bonferroni correction (Fig 5B and S3 Table). Immediate and neighbouring genes within 10 kb each side of the significant SNPs were considered as candidates (S3 Table) [61]. Two SNPs on the upper arm of chromosome 4 fall in the *bHLH059* transcription factor gene from AtbHLH group XI[62], in which a T-DNA insertion in Col-0 led to slightly increased resistance to a virulent strain of the oomycete pathogen *Hyaloperonospora arabidopsidis*[63]. Since *bHLH059* knock-out mutants are not sterile (S10F Fig) [63] we use *bHLH059* here rather than its initial name *UNFERTILIZED EMBRYO SAC 12*. None of the six remaining significant SNPs considered by the GWAPP tool was in a gene related to SA biosynthesis/signalling, temperature responses, defence or cell death regulation (S3 Table) except for *At4g02600*, a homologue of barley *mildew resistance locus 1* (*ATMLO1*) [64]. However, *ATMLO1* expression was found to be specific to early development[64]. In the GWAS analysis, *A. thaliana* genes involved in thermosensory regulation, such as *PIF4*, *PhyB*, *NTL6* or genes controlling alternative histone H2A. Z recruitment[7,10,13,21], were not found to be associated with temperature-dependent SA regulation. Because Bonferroni is conservative, we extended the list of candidate genes in the

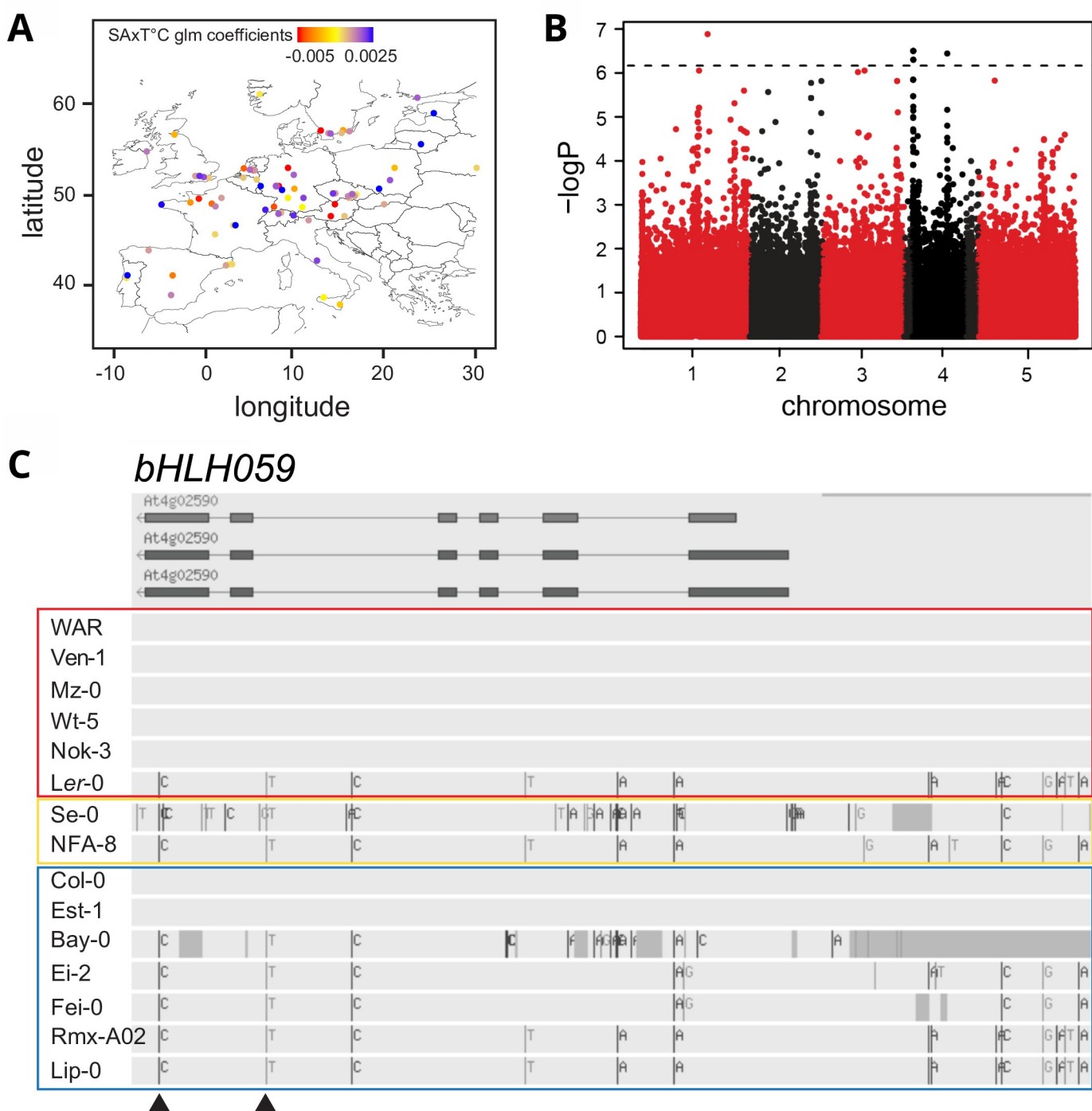

**Fig 5. Distribution and genetic architecture of SA regulation by temperature in *A. thaliana*. A**) Geographical distribution of 78 *A. thaliana* accessions from Europe, representing 75% of phenotyped accessions. Dot colours indicate phenotypes of SA regulation by temperature according to coefficients of our glm model. Colour scale represents accessions displaying higher SA contents at 22°C than at 16°C in red, equal SA contents in yellow and higher SA contents at 16°C than at 22°C in blue. **B**) Manhattan plot of association mapping with 99 *A. thaliana* accessions for SA regulation by temperature according to GWAPP using the coefficients of our glm model. Each dot represents a single nucleotide polymorphism and the dashed horizontal line indicates significant linkage disequilibrium threshold after Bonferroni multiple testing correction. **C**) *bHLH059* haplotypes according to Tair10 genome browser of accessions with extreme and intermediate total SA x T°C phenotypes. Colour blocks indicate phenogroups as displayed in phenotype distribution map in Fig 5A. Arrows indicate significant SNPs associated with total SA x T°C phenotype considered by GWAPP.

vicinity of SNPs with a reduced significance level of–log(P) = 5.5 (S3 Table). This identified SCF E3 ubiquitin ligase complex genes: *Skp1 interacting protein 5* (*SKIP5*), *cullin 1* (*CUL1*), two *F-Box protein* genes (*At3g25750*, *At3g54460*) and a *ubiquitin ligase protein degradation gene* (*At3g29270*) as candidates for temperature-dependent SA regulation (S3 Table). Several other candidates are associated with transcription (*bHLH059*, *RNA polymerase II E*, transcription factor *At2g46510* and transposable elements (*TE*) (S3 Table).

## Comparison of *bHLH059* functions with thermosensory immune regulator *PIF4*

As *bHLH059* is supported by two significantly associated SNPs on chromosome 4 (Fig 5B and 5C and S3 Table), we investigated its role in temperature-dependent SA accumulation and immunity. Using TAIR.10 sequence data to identify *bHLH059* genomic polymorphisms with the Col-0 reference genome (http://signal.salk.edu/atg1001/3.0/gebrowser.php) we examined sequences from accessions with extreme or intermediate temperature x SA phenotypes that were also used for the *Pst* DC3000 infection assays (Fig 3). Variation was uncovered in *bHLH059* coding and regulatory sequences (Fig 5C). While accessions Bay-0 and Se-0 have several deletions and nucleotide exchanges in the *bHLH059* coding sequence relative to Col-0, all other considered accessions only display one SNP in the coding sequence (Fig 5C). Since this SNP leads to a synonymous mutation, we reasoned that variation in expression of *bHLH059* rather than protein sequence might underlie temperature-modulated SA and/or bacterial resistance. Indeed, *bHLH059* expression measured in five key accessions revealed differential temperature regulation (S9 Fig). Whereas *bHLH059* expression was stable between 22°C and 16°C in Ven-1 and Mz-0 with higher SA accumulation at 22°C, it increased at 22°C in Fei-0, Ei-2 and Col-0 from the other phenogroup with reduced SA at 22°C (S9 Fig). Although *bHLH059* lacks a diagnostic SNP profile for its phenogroups in promoter and genic sequences (Fig 5C), the gene has a phenogroup expression pattern. Therefore, trans-regulatory elements might play a role in temperature-dependent *bHLH059* expression.

A T-DNA line with an insertion in the last intron of *bHLH059* leading to a truncated transcript (SALK_13303; *bhlh059-13* verified by qRT-PCR (Fig 6A and S4 Table)), and a β-estradiol inducible *bHLH059* transgenic line (*βE::bHLH059*) from the TRANSPLANTA Col-0 collection [65], were selected for phenotyping. Because the bHLH TF *PIF4* and its closest homologue *PIF5* are temperature-sensing immunity regulators under control of the PhyB thermosensory pathway in Col-0[10], we explored redundancy or cooperativity between *PIF4/PIF5* and *bHLH059* by including the *pif4-2 pif5-3* double mutant and a *PIF4::PIF4HA pif4-101* over-compensating transgenic line[66,67] in our assays. By performing a 2 way-ANOVA on expression of *BHLH059*, *PIF4* and *PR1*, SA contents and *Pst* DC3000 bacterial counts at 0 and 4 dpi in response to genotype and temperature in this set of mutant lines, we found that temperature significantly affects all our variables except *bHLH059* expression where it is marginally significant (p-value = 0.08) and bacterial counts at 0 dpi, in line with results shown in S5 Fig. We therefore tested for differences between mutant lines within each temperature regime to determine effects of *bHLH059* and *PIF4* misregulation on SA and bacterial resistance (Fig 6). In 5-week-old Col-0 plants at 16°C, *bHLH059* expression was low and increased 0.5-fold at 22°C (Fig 6A). As expected, *bhlh059-13* had no detectable full-length transcript whereas the estradiol-untreated *βE::bHLH059* line expressed 2-fold higher *bHLH059* than Col-0 at both temperatures (Fig 6A). *PIF4* expression was undetectable in *pif4-2 pif5-3* and was elevated in the *PIF4::PIF4HA pif4-101* line at 16°C compared to Col-0, and further boosted in *PIF4::PIF4HA pif4-101* leaves at 22°C (Fig 6B). Loss or gain of *PIF4* expression, respectively in *pif4-2 pif5-3* and *PIF4::PIF4HA pif4-101*, did not alter *bHLH059* expression at either temperature (Fig 6A). Reciprocally, *PIF4* expression which was

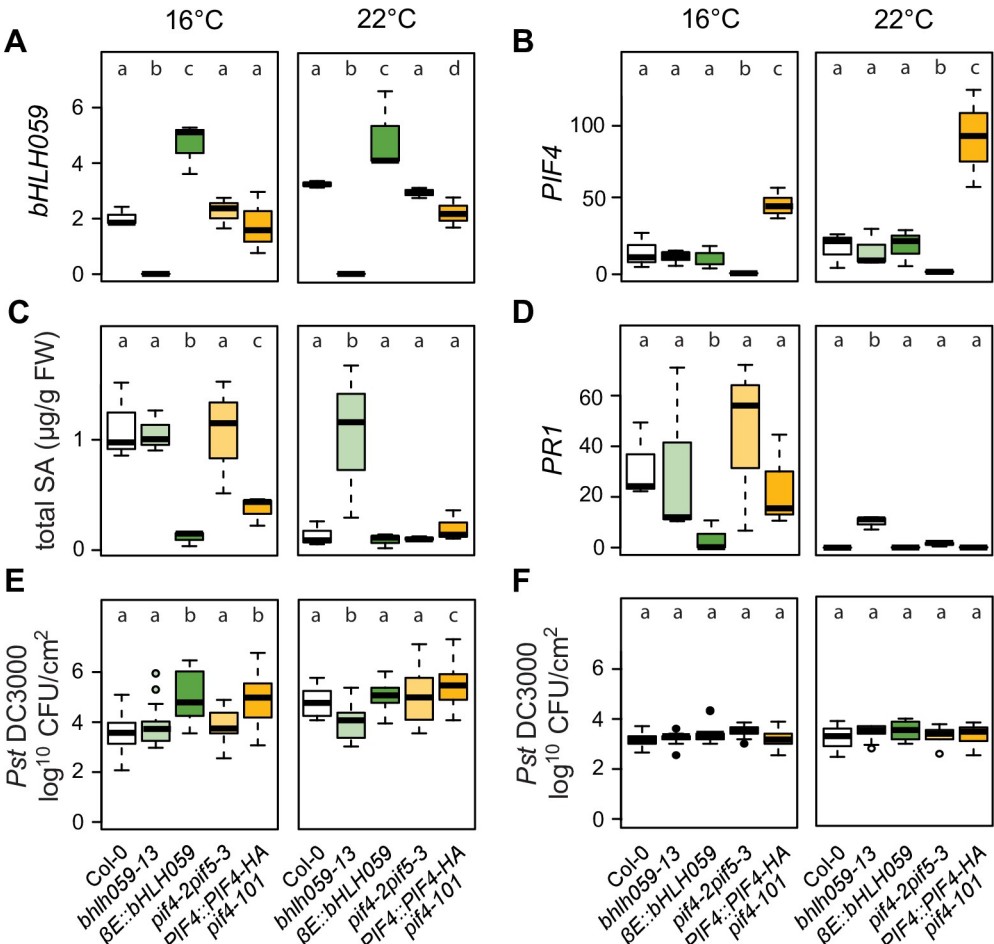

**Fig 6. Comparison of defence-related phenotypes in 5-week old Col-0 (white), *bHLH059* (green) or *PIF4* (orange) mutants and transgenic lines either grown at 16˚C or 22˚C.** Data are represented as boxplots. Letters represent significant differences between genotypes after Tukey's multiple testing correction in one way ANOVA. **A)** *bHLH059* expression relative to *SAND* reference gene in mature leaves of 5-week-old plants (n = 3 biological replicates). **B)** *PIF4* expression relative to *SAND* mature leaves of 5-week-old plants (n = 3 biological replicates). **C)** Total SA contents in mature leaves of 5-week-old plants (n = 3 biological replicates). **D)** *PR1* expression relative to *SAND* in mature leaves of 5-week-old plants (n = 3 biological replicates). **E)** *Pst* DC3000 growth in leaves at 4 d after spray inoculation (n = 18 from 3 biological replicates). **F)** *Pst* DC3000 initial titres in leaves 4 h after spray inoculation (n = 9 from 3 biological replicates).

higher than *bHLH059*, did not change within the 6˚C temperature range in Col-0 or *bhlh059-13* and *βE::bHLH059* lines (Fig 6B). Therefore, *bHLH059* and *PIF4/PIF5* do not influence each other's expression under the tested conditions.

## *bHLH059* has features of a temperature-responsive immunity regulator

We quantified total SA in the above lines at 16˚C and 22˚C. Col-0 SA levels decreased with increased temperature (Fig 6C) as observed before (Fig 3A). At 16˚C, the *bhlh059-13* mutant had similar total SA amounts as Col-0 but, unlike Col-0, maintained the same SA level at 22˚C (Fig 6C). Strikingly, 2-fold over-expression of *bHLH059* in the *βE::bHLH059* line led to low total SA accumulation at both temperatures (Fig 6C). Therefore, mis-regulation of *bHLH059* alters SA accumulation in response to temperature, with higher *bHLH059* expression dampening SA levels, similar to the behaviour of accessions Fei-0 and Ei-2 (Figs 3A and S9). We found

that *pif4-2 pif5-3* did not alter temperature modulation of total SA but that *PIF4* over expression (in *PIF4::PIF4HA pif4-101*) reduced total SA in plants grown at 16˚C and further at 22˚C (Fig 6C). These data suggest that *bHLH059* and *PIF4* operate differently in transmitting temperature information to SA accumulation.

Next we tested whether the temperature x SA profiles in the above lines tally with changes in SA-based immunity by quantifying *PR1* expression and *Pst* DC3000 growth in 5-week-old plants at 16˚C and 22˚C. *PR1* expression and resistance to *Pst* DC3000 correlated with SA regulation by temperature in these lines with two exceptions (Fig 6C–6E). At 22˚C, *PR1* expression in *bHLH059-13* was lower than expected based on its SA accumulation and resistance to *Pst* DC3000 (Fig 6C–6E). By contrast, the *PIF4* overexpression line (*PIF4::PIF4HA pif4-101*) exhibited equivalent low SA and *PR1* expression but higher *Pst* DC3000 susceptibility than Col-0 at 22˚C (Fig 6C–6E). At 3 hpi, *Pst* DC3000 bacterial entry was similar between temperature regimes and lines and neither genotype nor temperature had significant influence (Fig 6F), suggesting that the observed *bHLH059* and *PIF4* effects on bacterial resistance are mainly after initial stomatal entry. Taken together, these data suggest that *bHLH059* participates in temperature regulation of SA immunity and do not support a conjunction of *bHLH059* and *PIF4* pathways in this temperature response.

## bHLH059 mis-regulation does not alter *A. thaliana* development

*A. thaliana PIF4* has an important role in thermomorphogenesis in which it negatively regulates pathogen immunity to favour plant growth at higher temperature[6,10]. Mis-regulation of *PIF4* and its homologues alters hypocotyl and petiole elongation, growth and onset of flowering. We therefore compared developmental phenotypes of the *bhlh059-13*, *βE::bHLH059*, *pif4-2 pif5-3*, *PIF4::PIF4HA pif4-101* lines and Col-0 after 5 weeks at 16˚C and 22˚C and a 12 h light/dark cycle, as in the previous assays. The *bhlh059-13* mutant and *βE::bHLH059* over expression line resembled Col-0 in stature (Fig 7A). By contrast, *pif4-2 pif5-3* plants were stunted at 22˚C and *PIF4::PIF4HA pif4-101* had longer petioles (Fig 7A), as reported[10,68]. Hypocotyl lengths were similar between *bHLH059* lines and Col-0 at both temperatures, but were shorter in *pif4-2 pif5-3* and longer in *PIF4::PIF4HA pif4-101* at 16˚C and 22˚C (Fig 7B) [6]. Deviations from Col-0 were also observed in above-ground FWs of the tested *PIF4* lines at 16˚C and 22˚C, but not in the *bHLH059* lines (Fig 7C). Flowering time was similar in all lines at 16˚C but delayed in the *PIF4* lines at 22˚C (Fig 7D). Delayed flowering was reported for *PIF*-deficient lines grown at ~22˚C in short day conditions, whereas *PIF* over-expressors accelerated flowering[68,69]. Why this latter trend is not observed under our growth conditions remains unclear. Phenotypic analysis of an independent *bHLH059* T-DNA insertion line (SALK_010825C; *bhlh059-01*) [63] compared to wild-type Col-0 showed that it behaves similarly to *bhlh059-13* at 16˚C and 22˚C at the level of *bHLH059* and *PR1* expression, total SA accumulation and plant biomass (S10A–S10E Fig). These data lead us to conclude that *bHLH059* expression in *A. thaliana* Col-0 impacts temperature modulation of SA immunity without markedly altering developmental traits and that *bHLH059*-related thermosensory processes affecting SA accumulation and immunity to *Pst* DC3000 are distinct from those controlled by *PIF4*. Interestingly, all lines (S7B–S7E Fig) except Ven-1 (S7A Fig) show significant changes in bacterial titres as early as 1 dpi, emphasizing the strength of temperature-driven defence regulation.

## Discussion

Here we explored *A. thaliana* natural variation in response to temperature impacting SA accumulation, growth and resistance to bacterial (*Pst* DC3000) infection. One aim was to

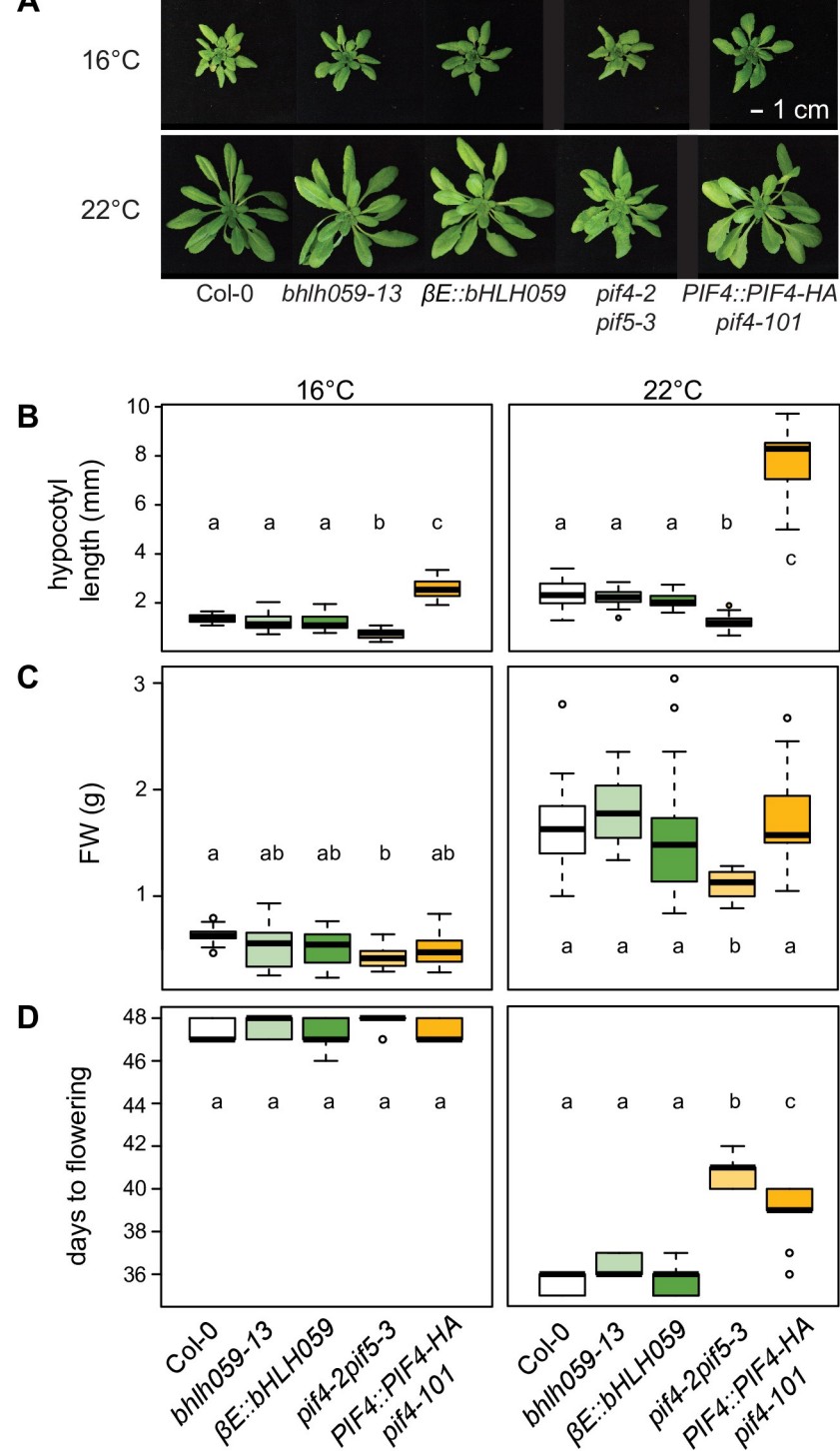

**Fig 7. Comparing developmental phenotypes of Col-0 (white), *bHLH059* (green) and *PIF4* lines (orange), as indicated, grown at 16˚C or 22˚C.** Letters represent significant differences between genotypes after Tukey's multiple testing correction in one-way ANOVA. **A)** Visual phenotypes of 5-week-old plants. **B)** Hypocotyl lengths of seedlings 10 d after germination (n = 15 from three biological replicates). **C)** Above-ground fresh weights of 5-week-old plants (n = 12 from three biological replicates). **D)** Days to flowering (n = 9 from three biological replicates).

determine differential temperature effects on immunity within a non-stress range, taking SA as an initial proxy for plant defence status. A second aim was to identify potential benefits and costs of accumulating high or low SA at a particular temperature. By testing 105 genetically diverse *A. thaliana* accessions, we uncover variation in total leaf SA accumulation between the 16˚C and 22˚C temperature regimes. We establish that increased SA amounts do not always correlate with reduced biomass, indicating a capacity of certain *A. thaliana* genotypes to mitigate negative effects of high SA levels on growth. Using a set of 11 selected accessions covering the range of observed temperature-modulated SA and growth responses, we detect a robust positive relationship between total SA in leaves prior to infection and restriction of *Pst* DC3000 growth, representing a possible benefit of accumulating SA. From an association study of temperature x SA in 99 of the 105 accessions, we identify *bHLH059* as a strong candidate for thermoresponsive control of SA immunity to *Pst* DC3000. This analysis uncovers diversity in plant responses to temperature and a way forward to understand the genetic architecture of plant adaptation to changing environments.

We were able to group *A. thaliana* accessions into three broad classes based on increased, decreased or stable total SA contents associated with the 16˚C—22˚C temperature difference (Figs 1C and 3A). *A. thaliana* Col-0, the most studied accession for temperature effects on immunity and growth[16,17,26,27,30,70], showed a comparatively weak negative SA accumulation trend with increased temperature (Figs 3A and 6C). In a previous study, higher SA in Col-0 plants grown at 5˚C compared to 23˚C contributed to growth retardation at the chilling temperature[27], consistent with tradeoffs between induced plant defences and growth[35,38].

*A. thaliana* autoimmunity phenotypes leading to growth inhibition and necrosis have been linked to increased SA[5,20,33]. It is therefore striking that ~ 33% of the 105 accessions with >1μg/g FW total SA retained a biomass above the median of the tested genotypes (Fig 2A and 2B). That certain accessions with high SA exhibited increased resistance to *Pst* DC3000 or *PR1* expression without a measurable biomass penalty (eg. Ven-1, Kas-2, PHW-13 in Figs 2C–2E and 3A and 3B), points to genotypic variation in the threshold at which SA leads to autoimmunity[28]. Presence of genetic modifiers of SA-related autoimmunity and stress sensitivity are evident from studies of different *A. thaliana* accessions[41,71]. Alterations in the hormone network controlling SA crosstalk with growth-promoting pathways might buffer against SA negative effects in some genotypes[72]. Notably, SA signalling contributes positively to petiole elongation in the *A. thaliana* Col-0 shade avoidance growth response [66]. Also, defences and growth were effectively uncoupled in *A. thaliana* Col-0 in a *Jasmonate-Zim Domain* (*JAZ*) repressor x *phyB* sextuple mutant, indicating that perturbation of the hormone transcriptional network can reduce defence-growth tradeoffs [43]. *A. thaliana* accession C24 displays an unusual broad-ranging tolerance to stress encounters with little negative impact on growth [73]. For accessions with different SA-growth relationships identified in our analysis (Fig 2C and 2D), it will be interesting in future studies to pinpoint underlying stress network properties and whether the apparent benefit of high SA on bacterial resistance creates vulnerabilities to other environmental stresses or conditions.

Phenotypic characterization of 11 differential accessions revealed a positive relationship between total SA accumulation in response to temperature and post-stomatal restriction of *Pst* DC3000 growth in leaves, with plants being more resistant at the temperature the respective unchallenged accession accumulated higher SA (Fig 3). SA amounts in plants within the studied 6˚C temperature range might thus be a predictor of basal resistance strength. Some accessions (eg. Ven-1, Nok-3, Ler-0, Fei-0) displayed more than 50-fold differences in *Pst* DC3000 growth at 4 dpi between the two temperatures (Fig 3B) which is similar to the differential growth observed between virulent and avirulent *Pst* strains in leaves of *A. thaliana* accessions Col-0 or Ws-2 [70]. Therefore, even a moderate temperature change can have a similar impact

on bacterial infection as pathogen effector-triggered immunity. Quantifying *Pst* DC3000 titres in the corresponding *NahG*-transgenic lines showed that SA depletion abolished the temperature effect on *Pst* DC3000 growth in all of the tested accessions (Fig 3). Hence, the temperature effect on resistance to bacterial growth appears to be an SA-dependent trait in these accessions. Huot et al (2017) established that SA signalling represents a major temperature-sensitive resistance node in *A. thaliana* accession Col-0, assessed over a warmer temperature range of 23–30˚C, and that high temperature suppression of immunity to *Pst* DC3000 was independent of *PhyB* and *PIF4*[16]. These and our data emphasize the importance of temperature differences within the normal range experienced by *A. thaliana* on effectiveness of SA-based pathogen immunity. In *Pst* DC3000-inoculated or *A. thaliana* autoimmune backgrounds, increased SA accumulation at 20–22˚C compared to 28–30˚C was associated with higher expression of the *ICS1* and *PBS3* SA biosynthesis genes [15,35,70]. By contrast, we found that changes in basal SA levels between 16˚C and 22˚C were not accompanied by altered *ICS1* or *PBS3* expression in five *A. thaliana* accessions (S4 Fig), suggesting that basal SA accumulation is controlled by another mechanism across this moderate temperature range.

The differences in *Pst* DC3000 titres remaining between *NahG*-expressing accessions (Fig 3C), expose a contribution of SA-independent processes to variation in resistance to virulent bacteria, as also indicated by a screen of 1041 *A. thaliana* accessions in response to spray-inoculated *Pst* DC3000[74]. Mz-0, our most resistant accession, displays an autoimmune phenotype at 22˚C with chlorosis, stunting and high SA accumulation due to a hyperactive allele at the *ACD6* locus that is also present in Est-1[49]. Nevertheless, the Mz-0 *NahG* line retained strong SA-independent resistance to *Pst* DC3000 (Fig 3C). In Col-0, post-stomatal basal and effector-triggered immunity to *Pst* DC3000 strains can be divided into parallel SA-dependent and SA-independent resistance branches [56,75,76]. SA-independent resistance provides some protection against pathogens that can disable SA pathways. Our and other analyses suggest that part of that resilience might lie at the level of maintaining SA-independent immunity over a range of temperatures[17,19].

The differences in SA contents regulated by temperature between *A. thaliana* accessions were not linked to a geographical distribution pattern (Fig 5A) and therefore it is not known whether this represents an adaptive trait to local climatic conditions[77]. Loci strongly associated with climate variables were enriched in amino acid-changing SNPs, indicating the presence of adaptive alleles[2]. There is increasing evidence for microhabitat effects such as edaphic conditions, intraspecific competition, herbivore distribution and altitude playing roles in local adaptation[3,78,79]. A study on the genetic basis to local adaptation of *A. thaliana* in Europe revealed several associated loci related to immunity[77]. Also, defence and cold response processes were associated with adaptive climate variables[2]. Taken together, these data suggest that temperature modulation of plant defences impact local adaptation.

Association mapping allowed us to link variation in temperature-dependent total SA to three loci on two chromosomes, with a strongly supported QTL on chromosome 4 (Fig 5B). In *A. thaliana* Col-0, *bHLH059* has features of a thermoresponsive immunity component because loss or mild over-expression of this gene disturbed temperature effects on SA accumulation and basal resistance to *Pst* DC3000 bacteria (Fig 6E). Interestingly, bHLH059 was identified as a weak negative component of Col-0 immunity and, in yeast 2-hybrid assays, as a potential defence hub connected to multiple NLRs[63].

Our comparative physiology and immunity phenotyping of *PIF4*/5 and *bHLH059* mis-expressed lines (Figs 6 and 7) suggest that these factors act independently in transmitting or processing temperature stimuli to immune and growth responses. Also, *bHLH059* was not identified as a PIF4 transcriptional target[80]. Therefore, in line with Huot et al (2017) findings, we think it unlikely that temperature modulation of *A. thaliana* SA defences involves

PIF4/5 signalling. Loss-of-function *bHLH059* mutations in accession Col-0 enhanced resistance to *Pst* DC3000 without an obvious physiological or developmental cost at 22˚C (Figs 6E and 7). It will be interesting to test if manipulating expression of *bHLH059* and/or its paralogue *bHLH007* (At1g03040)(61) in *A. thaliana* increases survival against pathogen infection over a range of temperatures.

## Methods

### Materials

For the temperature screen we used a sub-collection of 105 *A. thaliana* accessions from the Hapmap population (http://bergelson.uchicago.edu/wp-content/uploads/2015/04/Justins-360-lines.xls), provided by Maarten Koornneef (MPI for Plant Breeding Research, Cologne). This population was developed from a global collection of 5810 accessions to reduce redundancy and relatedness, which is a problem in GWA studies[81,82]. Accessions were chosen based on geographic distance and seed availability (S1 Table). No lines required vernalization to flower. *A. thaliana* transgenic lines used were: *sid2-1*[24], Col-0 *NahG*[58], Ler-0 *NahG*[83], Est-1 *NahG* and plasmid MT363 with the *NahG* construct[49] were provided by Detlef Weigel (MPI for Developmental Biology, Tübingen). Fei-0, Ei-2, Bay-0, Ven-1, Mz-0 and Nok-3 accessions were transformed with pMT363 via floral dipping as described[49]. *BHLH059* T-DNA insertion lines SALK_010825C (*bHLH059-01*) and SALK_13303 (*bHLH059-13*), and a β-estradiol-inducible line[65] were obtained from Nottingham Arabidopsis Stock Centre (http://nasc.nott.ac.uk) and *BHLH059* expression checked via RT-qPCR (S3 Table). Lines *pif4-2 pif5-3*[66] and *PIF4::PIF4HA pif4-101*[67] were provided by Christian Fankhauser (University of Lausanne).

### Plant growth conditions

*A.thaliana* plants were grown under controlled conditions at 16±1˚C (day) and 14±1˚C (night) or 22±1˚C (day) and 20±1˚C (night), 60±10% relative humidity, 200μE $m^2$ $s^{-1}$ light intensity and 12h day/night cycle. Seeds were first stratified in soil at 4˚C for 3 d. Plants were grown in individual 0.8l pots with commercial potting soil pretreated with entomopathogenic nematodes. Pots were distributed into trays in a fully randomized design for temperature x SA screening. One plant per accession was grown in each replicate. Data from three independent experiments (biological replicates) at 22˚C and four at 16˚C were used for analysis. Plants were grown in parallel in two growth chambers after ensuring replicability of results between chambers.

### Bacterial infection and *A. thaliana* physiology assays

For *Pst* DC3000 infection assays and *bHLH059/pif4* mutant characterization, plants were distributed in trays in a randomized design by genotype row. SA and gene expression assays were performed on three plants for each genotype in three independent experiments. Bacterial early entry into leaves through stomata was determined by measuring *in planta* bacterial titers at 3 hpi in a total of nine plants (with three individual plants per independent experiment). At 4 dpi, bacterial growth was determined in a total of 18 plants (derived from three independent experiments). Random groups of six genotypes were tested in parallel at each temperature with at least one wild type in each group. Trays were distributed randomly in a phytotron growth chamber and moved once a week to a new position. For spray-inoculation of 5-week-old plants with *Pst* DC3000, bacterial suspensions at 0.15 $OD_{600}$ in 10 mM $MgCl_2$ were used, as described[84]. *Pst* DC3000 infiltration into leaves was performed using a concentration of 0.0005 $OD_{600}$ injected with a flat-topped 1ml syringe.

## Bacterial growth in culture

Growth of *Pst* DC3000 (empty vector pVSP61, used for all *in planta* experiments) was assessed in 20 ml M9 minimal salt medium (per L: 100ml 10 x M9 salts, 100 μl 1 M CaCL2, 1000 μl 1M MgSO4, 25 g Sorbitol, 5g Sucrose pH 7.2 (NaOH)) with Rifampicin 40 μg/ ml, Kanamycin 25 μg/ ml, after transferring 200 ul of a 20 ml overnight culture in 5 ml LB medium (Rif 40 μg/ ml, Kan 25 μg/ml). Bacteria were then grown in the dark with shaking at 200 rpm for 56 h at 16˚C or 22˚C and $OD_{600}$ was measured with a photometer.

## Salicylic acid measurements

Luminescence produced by induction of an SA degradation operon coupled to a LUX cassette was used to measure SA in leaves using the biosensor-based method, as described[47,48]. Total and free SA was quantified by GC-MS as described[85]. For both methods, 100–200 mg leaf samples were frozen in liquid nitrogen and disrupted by a tissue lyzer (Retsch). Material was suspended in 250μl NaOAc 0.1M pH5.5 and mixed. Samples were centrifuged in a micro-fuge for 15 min, 200μl supernatants transferred to a 96 well PCR plate and treated with 4U of almond beta-glucosidase (Sigma) at 37˚C for 1.5 h. For bio-sensor measurements, 30μl sample was transferred to a 96 well black optiplate (Perkin Elmer) containing 60μl LB medium. A standard curve for SA (Sigma) was used in a volume of 10μl complemented with 20μl β-gluco-sidase-treated leaf extract of Col-0 *NahG* leaves to mimic leaf samples. The standard curve was designed and tested to measure SA concentrations from 0 to 20 μg total SA/g leaf fresh weight. The transgenic *Acinetobacter* luminescent strain was grown as described[85] and 50 μl bacte-rial suspensions ($OD_{600}$ 0.4) were added to each optiplate sample before incubating at 37˚C for 1h. The optiplate was then read by a luminometer (Berthold technologies) measuring lumines-cence emitted by each sample in 1/3s. Mean luminescence taken from three plate readings was used to calculate total SA concentrations. Because luminescence increase in the standard curve was non-linear we interpolated the data points using the approxfun() function in R (Cran 3.2.2.).

Gas chromatography-mass spectrometry (GC-MS)-based analyses of plant metabolites (SA and SAG) (S4A Fig) was performed as detailed in Hartmann et al 2018[86], with modifica-tions. 50 mg of pulverized, frozen leaf samples were extracted twice with 1 ml of MeOH/50 mM NaPO4 pH 6 in $H_2O$ (80:20, v/v). For internal standardization, 1 μg of D9-Pip, D9-NHP, D5-SA, indol-3-propionic acid (IPA), salicin, ribitol were added. 600 μl of the extract were evaporated to dryness. Free hydroxyl groups of the analytes were converted into their tri-methylsilyl derivatives by adding 20 μl of pyridine, 20 μl of N-methyl-N-trimethylsilyltrifluor-oacetamide (MSTFA) containing 1% trimethylchlorosilane (v/v) and 60 μl of hexane. The solution was heated to 70˚C for 30 min and after cooling, samples were diluted with hexane, and 2 μl of the solution was separated on a gas chromatograph (GC 7890A; Agilent Technolo-gies) equipped with a Phenomenex ZB-35 (30m x 0.25mm x 0.25μm) capillary column. The following GC temperature settings were used: 70˚C for 2 min, with 10˚C / min to 320˚C, 320˚C for 5 min. Mass spectra were recorded in the electron ionization mode between m/z 50 and m/z 750 with a 5975C mass spectrometric detector (Agilent Technologies). Metabolites were analyzed using the Agilent MSD ChemStation software. For quantification, substance peaks of selected ion chromatograms.

## Genome wide association study (GWAS)

Broad sense heritability of total SA contents in both temperature environments was estimated as $H^2 = \sigma^2 accession/ \sigma^2 accession+ \sigma^2 residuals$ based on a linear model using $\log^{10}$-transformed total SA data for normalization. Variation in temperature-dependent total SA was expressed

using the coefficients of the glm model: Total SA~accession*temperature+replicate fitted with a gamma distribution to account for variation due to temperature as well as between biological replicates. Association mapping was performed on 99 accessions using the GWAPP online application https://gwas.gmi.oeaw.ac.at/ [59]. We chose to further normalize the glm coefficients with a box cox power transformation and used the accelerated mixed model (AMM) to account for population structure[59]. SNPs were considered as significantly-associated with phenotype when they withstood Bonferroni multiple-testing correction.

## Quantitative RT-qPCR

RNA was extracted from liquid Nitrogen-frozen plant material using a my-budget Plant RNA kit (Bio Budget technologies Gmbh) according to manufacturer's instructions. cDNA was synthesized from 1µg plant RNA using M-MLV reverse transcriptase (Promega) following the manufacturer's protocol. RT-qPCR was performed with IQ SYBR Green supermix (Bio Rad) on a CFX Connect Real time system (Bio Rad). RT-qPCR primer sequences are listed in S3 Table. Relative expression of test genes was measured against *SAND* (*At2g28390*) as a stable reference gene[87]. Relative expression of genes was assessed in three plants per genotype, each from an independent experiment.

## Statistical analysis

We performed statistical analyses on at least three samples from three biological replicates using R package version 3.3.1. Comparison between two groups was done using a two-tailed Student t-test or non-parametric Kruskal Wallis test. Multiple comparisons were performed by one-way ANOVA and Tukey's multiple testing correction applied, except for data in Fig 2C–2E where false discovery rate was used. Raw data for Figs 1, 2A–2D, 3A and 5A are provided in S1 Table. For Fig 6 data, an initial 2 way ANOVA was performed. Correlations were assessed using Pearson's product moment correlation coefficient.

## Supporting information

**S1 Table. Phenotypic data from 105 A. thaliana accessions used to assess natural variation of biomass and total SA levels in response to temperature.**
(XLSX)

**S2 Table. Correlation between above ground fresh weight and dry weight in 5 week old A. thaliana accessions grown at 22˚C under controlled conditions.**
(XLSX)

**S3 Table. Genes in vicinity of SNPs highly associated with T˚C-dependent total SA homeostasis.**
(XLSX)

**S4 Table. Primer list for RT-qPCR.**
(XLSX)

**S1 Fig. Comparing *Acinetobacter* biosensor-based method and GC-MS analysis for measuring total and free SA contents in *A. thaliana* leaves of 7-week-old plants. A)** Total SA in four *A. thaliana* accessions with contrasting SA contents grown at 22˚C (n = 5 replicates from one experiment). Significant differences between methods after Student t-test with p-value<0.05 are indicated with stars on plot. NS = not significant. **B)** Free SA in four *A. thaliana* accessions with contrasting SA contents grown at 22˚C (n = 5 replicates from one experiment). Significant differences between methods after Student t-test with p-value<0.05 are

indicated on plot. NS = not significant.
(TIF)

**S2 Fig. Correlation of total SA and free SA contents measured by GC-MS in 15 *A*. *thaliana* accessions characterized in Figs 2, 3 and S3 in three biological replicates.** Plants were 5-week-old when sampled and grown at 22˚C. R = Pearson's correlation index (t = 14.365, df = 43, p-value $<$ 2.2e-16).
(TIF)

**S3 Fig. Total SA contents measured over developmental time in five *A*. *thaliana* accessions or mutants with contrasting SA contents (n = 12 from three biological replicates except for 15 d time point where n = 2).** Letters indicate significant differences after Tukey's multiple testing correction in one-way ANOVA. Circles indicate time point at which 100% plants were flowering. Grey shadows indicate standard error.
(TIF)

**S4 Fig. Regulation of temperature-dependent SA accumulation and relative expression of SA biosynthesis and defence genes using *SAND* as a reference gene in 5-week-old plants of the indicated accessions grown at 16˚C (black) and 22˚C (white). A)** Total SA in Ven-1, Mz-0, Fei-0, Ei-2 and Col-0 measured by GC-MS. Statistical differences according to Kruskal–Wallis rank sum test with p-value$<$0.05 are indicated with stars. N = 5 independent biological replicates except for Ei-2 where n = 3 **B)** *PR1*, **C)** *ICS1*, **D)** *PBS3*, and **E)** *PAL4* expression in Ven-1, Mz-0, Fei-0, Ei-2 and Col-0. Statistical differences according to Student t-test or Kruskal–Wallis rank sum test with p-value$<$0.05 between temperatures within each genotype are indicated with stars or NS = non significant on the graphic. N = 3 independent biological replicates.
(TIF)

**S5 Fig. Bacterial titres in leaves of *A*. *thaliana* accessions 4 h after spray inoculation with *Pst* DC3000. A)** Bacteria-inoculated 5-week-old plants of 11 *A*. *thaliana* accessions, as indicated, grown at 16˚C or 22˚C (n = 9, three biological replicates). Significant differences between temperatures after Student t-test with p-values $<$ 0.05 are indicated on plot with a star. NS = not significant. Error bars represent standard error. **B)** Bacteria-inoculated 5-week-old plants of 10 SA-deficient *A*. *thaliana* accessions grown at 16˚C or 22˚C (n = 9 from 3 biological replicates except for Ven-1 where n = 6). Significant differences between temperatures after Student t-test with p-values $<$ 0.05 are indicated on plot with a star. NS = not significant. Error bars represent standard error.
(TIF)

**S6 Fig. *Pst* DC3000 growth time course on minimal liquid medium.** Bacteria were measured by optical density ($OD_{600}$) increase over 56 h in M9 minimal salt medium with sorbitol at 16˚C (black) and 22˚C (white) (n = 3 from three biological replicates). Significant differences after Student t-test with p-value$<$0.05 are represented with stars. NS = not significant.
(TIF)

**S7 Fig. Growth time course over 4 days post infiltration (DPI) of *Pst* DC3000 at 16˚C and 22˚C in 5 *A*. *thaliana* ecotypes. A)** Ven-1 **B)** Mz-0 **C)** Se-0 **D)** Fei-0 **E)** Col-0. N = 12 for each time point by temperature and by genotype including four replicates for each of three independent biological replicates. Significant differences between temperature regimes at each time point with p-value$<$0.05 are indicate with stars on the graphic. NS = not significant.
(TIF)

**S8 Fig. Leaf total SA contents in 5-week-old *A. thaliana* accessions (black) and transgenic *A. thaliana* accessions transformed with a bacterial *NahG* gene (white).** Transgenic lines are homozygous except for Ven-1(due to a long generation time) for which two heterozygous lines were tested.Lines were phenotyped in the environment in which the parental line displayed highest SA accumulation to ensure full SA depletion (n = 3 from 3 biological replicates). Error bars represent standard error.
(TIF)

**S9 Fig. Temperature-dependent expression of *bHLH059* in ecotypes Ven-1, Mz-0, Fei-0, Ei-2 and Col-0 (n = 3 from independent biological replicates).** Differential expression in response to temperature according to student t-test with p-value<0.05 is indicated with stars on graphic. NS = not significant.
(TIF)

**S10 Fig. Comparison of phenotypes between Col-0 and *bhlh059-01* T-DNA insertion line in 5- week-old plants grown at 16°C and 22°C.** Data are represented as boxplots. Significant differences after student t-test with p-value<0.05 are represented with a star. NS = not significant. **A)** *bHLH059* expression relative to *SAND* reference gene in mature leaves (n = 3 from biological replicates). **B)** Total SA contents in mature leaves (n = 3 from biological replicates). **C)** *PR1* expression levels relative to *SAND* in mature leaves (n = 3 from biological replicates). **D)** Above-ground fresh weight (n = 3 from 3 biological replicates). **e)** Visual phenotypes of lines at 16°C and 22°C. **F)** Inflorescence with mature siliques of Col-0 and *bHLH059* mutant lines.
(TIF)

## Acknowledgments

We thank Jonas Klasen, Dmitry Lapin and Rubén Garrido-Oter for help with statistics and Angela Hancock (MPI for Plant Breeding Research, Cologne) and Rubén Alcázar (University of Barcelona) for constructive comments on the manuscript.

## Author Contributions

**Conceptualization:** Friederike Bruessow, Jane E. Parker.

**Data curation:** Friederike Bruessow, Jane E. Parker.

**Formal analysis:** Friederike Bruessow.

**Funding acquisition:** Friederike Bruessow, Jane E. Parker.

**Investigation:** Friederike Bruessow, Jaqueline Bautor, Gesa Hoffmann, Ipek Yildiz.

**Methodology:** Friederike Bruessow, Jane E. Parker.

**Project administration:** Jane E. Parker.

**Resources:** Jürgen Zeier, Jane E. Parker.

**Software:** Friederike Bruessow.

**Supervision:** Friederike Bruessow, Jane E. Parker.

**Validation:** Friederike Bruessow.

**Visualization:** Friederike Bruessow.

**Writing – original draft:** Friederike Bruessow, Jane E. Parker.

**Writing – review & editing:** Friederike Bruessow, Jane E. Parker.

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
