## [Decision Letter · Decision Letter 0]

12 Feb 2020

Dear Dr Parker,

Thank you very much for submitting your Research Article entitled 'Arabidopsis thaliana natural variation in temperature-modulated immunity uncovers transcription factor UNE12 as a thermoresponsive regulator' to PLOS Genetics. Your manuscript was fully evaluated at the editorial level and by 2 independent peer reviewers.

The reviewers appreciated the attention to an important problem, but raised some substantial concerns. Reviewer 1 urges an experiment with direct apoplastic infiltration of the bacteria Pst DC3000, as well as an RT-qPCR experiment to examine expression of UNE12 and Reviewer 2 has concerns about the variance in measurements of SA. Based on the scope of those requests, perhaps this manuscript is beyond a "major" revision and might be rejected with an invitation to resubmit. However, given the enthusiasm for the work as a whole, we will ask the authors to address the concerns either through new experiments or through a detailed response to the reviewers that would have them agree that the initial concerns are not major.

Regardless, based on the reviews, we will not be able to accept this version of the manuscript, but we would be willing to review again a version addressing the reveiwers' concerns. We cannot, of course, promise publication at that time.

If you decide to revise the manuscript for further consideration at PLOS Genetics, please aim to resubmit within the next 60 days, unless it will take extra time to address the concerns of the reviewers, in which case we would appreciate an expected resubmission date by email to plosgenetics@plos.org.

[LINK]

We are sorry that we cannot be more positive about your manuscript at this stage. Please do not hesitate to contact us if you have any concerns or questions.

Yours sincerely,

Rodney Mauricio, Ph.D.

Associate Editor

PLOS Genetics

Gregory P. Copenhaver

Editor-in-Chief

PLOS Genetics

Reviewer's Responses to Questions

**Comments to the Authors:**

Reviewer #1: SUMMARY:

Bruessow and co-authors have conducted a comprehensive phenotyping of 105 diverse accessions of Arabidopsis thaliana in terms of basal salicylic acid (SA) levels and aboveground biomass at two non-stress temperatures of 16°C and 22°C. Taking advantage of the natural variation in the species, they found that, while several accessions follow the trend of the commonly used Col-0 (higher SA levels at 16°C vs. 22°C), some accessions have the opposite trend (higher SA levels at 22°C vs. 16°C).

Based on these phenogroups, GWAS analyses led to the identification of UNE12 as a temperature-associated regulator of SA accumulation. This study is both significant and novel, as it identifies a new transcription factor that could regulate an important temperature-dependent pathway, like SA production. Rich datasets are also provided for the research community in terms of growth and SA phenotypes of a large collection of Arabidopsis accession globally.

Remarkably, the authors additionally observed that some accessions are able to mitigate growth tradeoffs due to SA, which makes the observations in Col-0 not a generalizable phenomenon and provides further evidence to alleviation of growth-defense tradeoffs as shown in previous studies with phyB-jaz (Campos et al., 2016 Nat Commun 7) and uORF-mediated gene regulation (Xu et al., 2017 Nature 545). Additionally, regardless of accession and temperature, the authors demonstrate the indispensability of SA on resistance against the pathogen Pseudomonas syringae DC3000.

This study is limited by the range of temperature investigated (16°C vs 22°C), which does not cover the regulation of SA levels by cold temperatures (Kim et al., 2017 Plant Cell 29) and warm temperatures (Mang et al., 2012 Plant Cell 24). Also, more mechanistic details not present in the current paper could be provided and elucidated in future studies. Nonetheless, the research presented in this manuscript should be of general interest to a broad readership, as it is at the intersection of plant immunity, plant-environment interactions and plant natural variation.

To further improve the manuscript, there are some issues that the authors need to address as detailed in the next sections.

MAJOR ISSUES:

1) One key issue that the authors need to absolutely resolve is their exclusion of “stomatal resistance as contributing significantly to the temperature effects observed on bacterial infection in these 11 accessions” (Line 186-188). To properly delineate the stomatal vs. apoplastic contributions of the plant defense responses, the authors need to perform direct apoplastic infiltration of Pst DC3000 in Figures 3 and 6, and also show that the same trends prevail when compared to spray-inoculations. The 4h monitoring of Pst DC3000 levels after spraying (Figure S4) only represents a single timepoint and does not exclude the stomatal dynamics before or after that timepoint. Because the conclusion on post-stomatal resistance is at the crux of this paper, the infiltration experiments are crucial.

2) Another key issue that the authors need to validate is the lack of amino acid mutations in the UNE12 gene between the various accession phenogroups. As stated in lines 297-299, “Since this SNP leads to a synonymous mutation, we reasoned that variation in expression of UNE12 rather than protein sequence might underlie temperature-modulated SA and/or bacterial resistance.” To validate this conclusion and because the GWAS identification of UNE12 is the main thesis of this paper, the authors need to show differences in UNE12 gene expression (by RT-qPCR) between the various phenogroups of Arabidopsis accessions. In relation to this, the authors could provide details if there are also polymorphisms that lie within the UNE12 gene promoter and show if putative regulatory control elements are affected by the nucleotide change/s (if applicable).

3) Finally, to add additional mechanisms to the temperature-sensitive basal SA levels at 16°C vs. 22°C, the authors could demonstrate which metabolic pathway mediates this phenomenon. Mang et al. (2012 Plant Cell 24) showed that the temperature sensitivity of basal SA levels at 22°C vs. 28C is due to differences in basal ICS1 gene expression. Because the authors show that the difference between 16°C and 22°C is in the basal SA levels, it would be good to resolve if this is due to the ICS1/PBS3 pathway as well by measuring gene expression levels.

MINOR ISSUES:

Figure 5: Col-0 and Est-1 belong to one phenogroup (higher SA at 16°C than at 22°C), while Ven-1, Mz-0 and Nok-3 belong to another phenogroup (higher SA at 22°C than 22°C). However, they all do not have polymorphisms in UNE12. Can the authors provide possible explanations for this in the Discussion?

In Lines 439-441, the authors emphasize that Mz-0 has an hyperactive ACD6 allele. They could also mention that this is the case with Est-1, which belongs to another phenogroup.

Specific comments on Figures and Tables:

• Figure 7: It is mentioned that “n=14 from three biological replicates” – was one experiment missing a sample? Please clarify.

• Suppl Figure S1: Was this one independent experiment? Or was this experiment repeated?

• Suppl Figure S2: Which 15 accessions were used?

• Suppl Figure S6: What is the extra bar in Ven-1?

• Suppl Figure S7b and c: There are larger variations in SA and PR1 levels in the une12 mutant compared to those in Col-0. Are there samples that are almost outliers in the une12 measurements?

• Suppl Table S4: Please indicate primer annealing temperatures used.

Specific comments on the text:

• Line 2: A title suggestion could be “Natural variation in temperature-modulated immunity uncovers transcription factor UNE12 as a thermoresponsive regulator in Arabidopsis thaliana”

• Line 18: Rephrase “… endogenous pathways remains unclear….” to “… endogenous pathways is not fully characterized….” There have been various studies (i.e. PhyB, PIFs, H2A.Z and NLRs) showing how temperature intersects with endogenous pathways.

• Line 22: In the line “…limiting bacterial pathogen,” please specify the pathogen.

• Line 60: Please mention that temperature can intersect with the HSP90 proteins (Wang et al., 2015 Nat Commun 7), SIZ1 E3 ligase (Hammoudi et al., 2018 PLoS Genetics 14) and the DET1/COP1 photomorphogenic regulatory module (Gangappa and Kumar, 2018 Cell Rep 25)

• Line 70: Please mention that PTI can also be negative regulated by higher stress temperatures when measuring different defense outputs (Janda et al., 2019 Mol Plant Pathol 20)

• Line 79: Fix the citation number “19” as a superscript.

• Line 98: Change “… variation in immunity responses…” to “… variation in immunity and growth responses…”

• Line 118: “Biomass” is both singular and plural.

• Line 843: Change to “Manhattan plot”.

Reviewer #2: The manuscript titled “Arabidopsis thaliana natural variation in temperature-modulated immunity uncovers transcription factor UNE12 as a thermoresponsive regulator” by Bruessow et al. examined variations among 105 Arabidopsis thaliana natural accessions regarding their defense hormone salicylic acid (SA) levels and plant growth at two temperatures (22oC and 16oC). They observed a within-species plasticity in SA-growth tradeoffs, revealing that high SA does not necessarily leads to stunting morphology. Through GWAS analysis, a bHLH transcription factor unfertilized embryo sac 12 (UNE12) was identified for the regulation of SA-growth trade-off. It serves differently with a known thermosensitive coordinator of immunity and growth, PIF4/5.

As much of our current understanding of plant immunity and growth trade-offs are based on A. thaliana genetic accession Col-0, this study expands our knowledge on A. thaliana natural variation in SA accumulation at different temperatures. Such analysis has been overlooked in the past, which reveals an unexpected diversity of SA regulation. Most of the experiments are designed rationally and most data are convincing. The manuscript is well written and easy to read. This reviewer has the following suggestions for improving the quality of their data and strengthening their conclusions.

Major points:

1. Huge variation is observed in the SA level measurements in Table S1, some with >10-fold differences among repeats. Therefore it is critical to corroborate the important SA levels data for the selected ecotypes (in Fig 2/3/4/6) using the more accurate HPLC-based or other methods. This will reduce the variations and hopefully error bars to reveal the trends better. The authors should also indicate in the legends how many times the quantitative experiments are repeated.

2. Corresponding to Fig 2c/d/e/f which were collected at 22C, the authors should show respective data at 16C to strengthen their conclusions.

3. A graph can be added to Fig 2 to summarize Table S1 and support the authors’ conclusions on SA-growth tradeoff, with X-axis being ecotype names and Y-axis being the ratio of SA to fresh weight (SA/FW). The curve at 16C can be arranged to be incremental, while the curve at 22 would be random.

4. In the results section titled genetic variation in A. thaliana SA-growth tradeoffs, it is stated that Ven-1, PHW-13 and Kas-2 had high total SA amounts and high biomass. However, according to the data shown in Table S1, total SA amounts of PHW-13 was quite low (<0.3 μg/g FW) at both temperatures. Please explain why it can be classified into high SA, high biomass group. Also, in Fig. 2d, the total SA amount of Ven-1, PHW-13 and Kas-2 were comparable, which is not consistent with the data on Table S1, please explain.

5. In the results section titled genetic architecture of SA regulation by temperature, it is stated that 99 accessions were used to perform temperature x SA association analysis. Why not use all 105 accessions? Please explain.

6. In Fig. 6 and Fig.7, a UNE12 T-DNA insertion line, une12-13, was used to perform the assays. But in the end of results section, another T-DNA insertion line, une12-01, was mentioned. It is stated that these two lines behaved similarly. So, what’s the reason here using another T-DNA line? Please clarify.

7. Although UNE12 is the name associated with the bHLH TF studied, it is quite misleading as the mutants do not seem to exhibit lethality or sterility. The inherited UNE conclusion is likely incorrect as it was from an early large scale screen. I would suggest the authors to use bHLH59 instead (see Heim et al 2003), and just mentioned that it also carries the UNE12 name (it might be worthwhile to show in a supplementary figure that there is no obvious sterility/lethal phenotype as suggested by UNE, just to clarify the literature).

Minor points:

The figures can be annotated better to assist with reading. For example, In Fig 2a/2b, the ecotypes selected for careful analysis should be indicated in the plots. The colours of Fig 2c/d/e should be explained (green/grey/purple) in the legend.

Line 354 – Reference numbers should be superscript.

References section – The names of cited paper are not consistent. For instance, in line 586, all the initials are capitalized. But in line 590, only the initial of first word is capitalized.

**Have all data underlying the figures and results presented in the manuscript been provided?**

Reviewer #1: Yes

Reviewer #2: Yes

PLOS authors have the option to publish the peer review history of their article (what does this mean?). If published, this will include your full peer review and any attached files.

Reviewer #1: No

Reviewer #2: No

---

## [Decision Letter · Decision Letter 1]

23 Oct 2020

Dear Dr Parker,

Thank you very much for submitting your Research Article entitled 'Natural variation in temperature-modulated immunity uncovers transcription factor bHLH059 as a thermoresponsive regulator in Arabidopsis thaliana' to PLOS Genetics. Your manuscript was fully evaluated at the editorial level and by independent peer reviewers. The reviewers appreciated the attention to an important topic but identified some aspects of the manuscript that should be improved.

We therefore ask you to modify the manuscript according to the review recommendations before we can consider your manuscript for acceptance. Both reviewers and the associate editor raise questions about the results presented in figure 6. Reviewer 1 suggests that a 2 way ANOVA would be the most appropriate first analysis rather than 2 separate 1 way ANOVA. You should do that analysis. Both reviewers raised issues about the clarity of that figure as well and you should do what you can to amke that figure as clear as possible.

[LINK]

Yours sincerely,

Rodney Mauricio, Ph.D.

Associate Editor

PLOS Genetics

Gregory P. Copenhaver

Editor-in-Chief

PLOS Genetics

Reviewer's Responses to Questions

**Comments to the Authors:**

Reviewer #1: Dear Authors,

Thank you very much to the authors for submitting this revised version of their manuscript. Bruessow and co-authors have addressed my major concerns as a reviewer:

• They have performed pathogen infiltration experiments to properly delineate the stomatal vs. apoplastic contributions of the plant defenses and have shown similar trends between spray-inoculations and infiltrations.

• Because the various temp X SA phenogroups did not show bHLH059 sequence differences, the authors successfully demonstrated differences in UNE12 gene expression between the phenogroups as they had previously speculated in the initial manuscript version.

• The paper clarified that temperature sensitivity of the SA pathway in the 16°C-22°C range is not due to differences in ICS1/PBS3 gene expression as previously demonstrated by Mang et al., 2012 and Huot et al., 2017.

To polish this manuscript, there are a few minor issues that the authors need to address as outlined below:

• If another revision is requested by the Editors, please make sure to include line numbers as was the case in the first version.

• Introduction, second paragraph (page 3): It would be great to update your literature review with the recently identified ELF thermosensing mechanism (Jung et al., 2020; https://www.nature.com/articles/s41586-020-2644-7).

• “High SA accumulation..” section (page 10): Please qualify the concluding sentence of the paragraph by adding “in the 16°C-22°C range.”

• “SA underlies differential…” section (page 11): In your concluding sentence, you state “These data suggest that … that the isochorismate SA biosynthesis pathway is sensitive to the temperature range used here.” I suggest that this second clause be deleted because you did not see gene expression differences in ICS1 and PBS3.

• Figure 6: I just noticed that one-way ANOVA was performed separately for each temperature. To facilitate easier comparisons between the two temperature within each genotype, I believe a two-way ANOVA with Tukey’s HSD may be more appropriate. In its current version, the statistical presentation (of the mean separations as letters) can be a bit confusing. For example, in panels c, d and e – the Col-0 values at 16°C vs. 22°C are statistically significant with the student’s t-test but have the same ANOVA letters, while the bhlh059 mutant and estradiol lines have different ANOVA letters (comparing the two temperatures) but are not statistically different.

Reviewer #2: The authors addressed most of my concerns. One small suggestion:

For Fig 6E, please clarify the statistic analysis in the legend. I am a bit confused on the a/b/c categories, some were deemed different, but they seem quite similar to me. Please clarify.

**Have all data underlying the figures and results presented in the manuscript been provided?**

Reviewer #1: Yes

Reviewer #2: Yes

PLOS authors have the option to publish the peer review history of their article (what does this mean?). If published, this will include your full peer review and any attached files.

Reviewer #1: No

Reviewer #2: No

---

## [Editor Report · Decision Letter 2]

10 Nov 2020

Dear Dr Parker,

We are pleased to inform you that your manuscript entitled "Natural variation in temperature-modulated immunity uncovers transcription factor bHLH059 as a thermoresponsive regulator in Arabidopsis thaliana" has been editorially accepted for publication in PLOS Genetics. Congratulations!

Yours sincerely,

Rodney Mauricio, Ph.D.

Associate Editor

PLOS Genetics

Gregory P. Copenhaver

Editor-in-Chief

PLOS Genetics

Comments from the reviewers (if applicable):

**Data Deposition**

http://datadryad.org/submit?journalID=pgenetics&manu=PGENETICS-D-19-02017R2

**Press Queries**

---

## [Editor Report · Acceptance letter]

19 Jan 2021

PGENETICS-D-19-02017R2 

Natural variation in temperature-modulated immunity uncovers transcription factor bHLH059 as a thermoresponsive regulator in Arabidopsis thaliana 

Dear Dr Parker, 

We are pleased to inform you that your manuscript entitled "Natural variation in temperature-modulated immunity uncovers transcription factor bHLH059 as a thermoresponsive regulator in Arabidopsis thaliana" has been formally accepted for publication in PLOS Genetics! Your manuscript is now with our production department and you will be notified of the publication date in due course.

With kind regards,

Melanie Wincott

PLOS Genetics

On behalf of:
